# Development of Curriculum Design Support System Based on Word Embedding and Terminology Extraction

**HoSung Woo [1]** **, JaMee Kim [2] and WonGyu Lee [3],\***

[1] Department of Computer Science and Engineering, Graduate School, Korea University, Seoul 136-701, Korea; hosung.woo@inc.korea.ac.kr
[2] Major of Computer Science Education, Graduate School of Education, Korea University, Seoul 136-701, Korea; celine@korea.ac.kr
[3] Department of Computer Science and Engineering, College of Informatics, Korea University, Seoul 136-701, Korea
\* Correspondence: lee@inc.korea.ac.kr

**Abstract:** The principles of computer skills have been included in primary and secondary educated since the early 2000s, and the reform of curricula is related to the development of IT. Therefore, curricula should reflect the latest technological trends and needs of society. The development of a curriculum involves the subjective judgment of a few experts or professors to extract knowledge from several similar documents. More objective extraction needs to be based on standardized terminology, and professional terminology can help build content frames for organizing curricula. The purpose of this study is to develop a smart system for extracting terms from the body of computer science (CS) knowledge and organizing knowledge areas. The extracted terms are composed of semantically similar knowledge areas, using the word2vec model. We analyzed a higher-education CS standards document and compiled a dictionary of technical terms with a hierarchical clustering structure. Based on the developed terminology dictionary, a specialized system is proposed to enhance the efficiency and objectivity of terminology extraction. The analysis of high school education courses in India and Israel using the technical term extraction system found that (1) technical terms for Software Development Fundamentals were extracted at a high rate in entry-level courses, (2) in advanced courses, the ratio of technical terms in the areas of Architecture and Organization, Programming Languages, and Software Engineering areas was high, and (3) electives that deal with advanced content had a high percentage of technical terms related to information systems.

**Keywords:** computer science curriculum; body of knowledge; curriculum analysis; terminology extraction system; word embedding

---

## 1. Introduction

The development of IT, including virtual reality, block chain using P2P networking, and intelligent ones based on learning algorithms, smartly solves transportation problems, environmental problems, and inefficiencies in facilities [1–4]. Social change due to rapidly changing technology is affecting elementary and middle school computer science (CS) education. Since the early 2000s, elementary and secondary education has included the principles of CS. Starting in 2013, there has been a reorganization effort to reflect social changes in technology for education in India, the United Kingdom, Japan, the United States, and Korea. Rather than reflecting only the latest technological trends [5], we are pursuing education that can enhance our cognitive ability to cope with unfamiliar problems.

The composition of any curriculum reflects knowledge selectively based on contents that experts in a given field think are necessary. This method of relying experts is not only costly and time consuming but also has a limitation in that if the composition selected by the experts changes, the curriculum becomes inconsistent, as contents of the curriculum may reflect the subjective process of selection.

It is not necessary to concentrate on educating IT specialists but rather on educating future members of the IT industry on the basic competencies they should have [6]. It is necessary to construct an objective and systematic curriculum based on contents that provide the competency needed for students to realize their potential. Though the education standards of different countries are based on the same knowledge system, the contents of curricula in the field of CS will differ between countries. As education reflects cultural characteristics, it can be said that education is affected by the social aspects of each country [7].

It is necessary to reflect the global trend in elementary and middle school education for CS, as it is a special field that rapidly drives forward technological change and society. For this reason, such education should be based on an objective flow, such as frame and knowledge extraction for curriculum development and curriculum analysis for each country. Rather than relying on a handful of experts in the extraction of knowledge, there is a need for a basis of standardized terminology to provide objective organization. Smart systems that support curriculum design will help ensure objectivity and efficiency.

This study focused on the extraction of terms and the composition of knowledge domains, which are content elements that help build a curriculum that reflects the standards of CS education and global trends. Our motivation is to develop a curriculum design support system (CDSS) that can help grasp the global trend in the curriculum development of elementary and secondary school.

This paper is organized as follows. Section 2 describes the research related to term extraction. Section 3 describes the terminology of CS based on curriculum standards. Section 4 describes the terminology dictionary construction and CDSS. In Section 5, conclusions are drawn from the results of the study, and the significance of this study is described along with directions for future research.

## 2. Related Work

### 2.1. Terminology Extraction

Indexers extract and construct important words from documents. However, jargon in a document may not be considered important, despite helping to understand a specific field [8]. Terminology in a specific field may have meaning only within that field, and some words may have different meanings depending on the context.

Term extraction is time-consuming and costly, and studies have been conducted to automatically extract terms. There are three broad categories of term extraction methods.

First, the statistical-based method extracts terms that satisfy a threshold value using statistical properties such as word frequency and term frequency-inverse document frequency, through models such as the hidden Markov model, maximum entropy model, and conditional random fields model. The statistical basis is advantageous because it is not affected by domain constraints and therefore is highly portable [9–11]. However, the low accuracy of extracted terms and the inclusion of noise affect the difficulty of meaning interpretation.

Second, the rule-based method recognizes and extracts terms using a general-purpose corpus that has been developed. A large number of candidate terms are analyzed and manually processed through morphemes such as prefixes and suffixes [12,13]. There is a resulting drawback of the researcher having low portability because they manually define and complement rules for each specific field [14]. In addition, because a general-purpose corpus is used rather than a corpus corresponding to a specific field, it is difficult to extract terms used in specialization fields with this method, and terms having a low frequency are not included.

Third, the hybrid method is a combination of statistical- and rule-based methods. The hybrid term extraction method extracts a string of a certain frequency as a candidate word and then uses a rule for parts of speech [15]. Although the hybrid method generally has better performance than the existing rule- and statistical-based methods, there is a disadvantage in terminology extraction, as its accuracy degrades when it is not supported by a corpus [16]. In the case of terms with high frequency, there is a disadvantage in that it will not extract an erroneous term or a term that has a low frequency [17]. However, the rule-based method is used in many studies because of its simple system implementation and high accuracy of extracted terms.

The terminology used in a curriculum can be linked to learning factors, achievement levels, and teaching and learning methods to determine the quality of education [18]. Since the accuracy of term extraction is therefore important, this study uses the rule-based method. In addition, we have developed a specialized corpus for the CS field and have applied it to the system to overcome the limitation of using a general-purpose corpus. Based on the extracted terms, the knowledge domain was constructed and the curriculum was designed.

*2.2. Word2vec*

Word2vec is a kind of word embedding technique in which words are expressed by vectors [19]. This technique is represented by vectors whose semantically similar words have a distribution of similar values, and it is used in various fields such as text classification [20] and sentiment classification [21].

Map sentences consisting of words that are not obviously related to each other to a higher-level matrix and replace semantic relationships between words with mathematical relationships in the matrix. Word2vec has two models: CBOW (continuous bag-of-words) and Skip-gram. This study describes Skip-gram with better performance than CBOW. Suppose a sentence in the corpus is S(Wt−n, Wt−n+l, … Wt,···,Wt+n−1,Wt+n), where Wi is a word. The structure of the skip-gram model is shown in Figure 1.

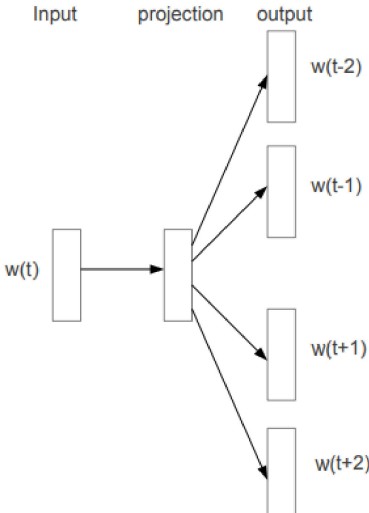

**Figure 1.** Skip-gram model architecture.

The Skip Gram model predicts a context vector through the word Wt. This means that Wt word vectors are superimposed on adjacent word vectors and are repeated several times before having a word vector with a similar context. In this study, Word2vec was used to organize the knowledge area of the curriculum.

## 3. Computer Science Terminology and Knowledge Areas

'Terminology' is a linguistic expression for a specific piece of knowledge and is used to convey meaning without misinterpretation. The exclusion of synonymous terms can be achieved by referring to physical data elements by the names of the corresponding logical data elements. The terminology dictionary used in Korea is the Information Communication Dictionary of the Telecommunications Technology Association, which was developed based on the CC2005 standard [22].

In the field of CS, the Association for Computing Machinery (ACM) and the Institute of Electrical and Electronics Engineers (IEEE) published the Guidelines for Computing Curriculum in 1968, laying the foundation for higher education. After its initial publication, updates to this curriculum guideline work have been published every 10 years. The first of the curriculum standards is CC2001. CC2001 influenced the composition of CC2005 as well as Japan's higher education standard, J07 [23]. CC2005 incorporates the IS2002, SE2004, CE2004, and IT2005, as well as specific disciplines of the CS field, in an effort to integrate the curriculum into the field of computation [22]. Thereafter, work on the curriculum standards for each field has been carried out. A curriculum for the CS field is proposed in the Computer Science Curricula 2013 (CS2013), which reflects the reorganization of CS2008 and the technological trends in 2013 [24,25]. CS2013 added 'information assurance and security', 'platform-based development', 'parallel and distributed computing', and 'system foundation' to reflect social needs and the latest technological trends. We have completed a knowledge system that reflects social and technological changes according to the weight of the knowledge domain, the creation of new knowledge domains, and the integration of existing knowledge domains. CS2013 has 18 areas of expertise at the higher education level, as shown in Table 1 [25].

**Table 1.** Computer Science Curricula 2013 (CS2013) knowledge area.

| | | |
|---|---|---|
| AL) Algorithms and Complexity | AR) Architecture and Organization | CN) Computational Science |
| DS) Discrete Structures | GV) Graphics and Visualization | HCI) Human-Computer Interaction |
| IAS) Information Assurance and Security | IM) Information Management | IS) Intelligent Systems |
| NC) Networking and Communication | OS) Operating System | PBD) Platform-based Development |
| PD) Parallel and Distributed Computing | PL) Programming Language | SDF) Software Development Fundamentals |
| SE) Software Engineering | SF) Systems Fundamentals | SP) Social Issues and Professional Practice |

Table 1 reflects a model curriculum for K–12 CS, published in 2003 and 2011 [26,27]. In the case of the CS field, because the curriculum is used not only in higher education but also from elementary to high school, it is necessary to consider a hierarchy in terms of the consistency and continuity of the curriculum.

In Japan, CC2001 is referred to as J07-CS for higher education [23]. The J07-minor, which can be used in this series, provides a curriculum based on jargon. A high school CS curriculum was formed in 2010 by simplifying the content of the higher education curriculum. When constructing a curriculum, it is necessary to consider not only the meaning of a term but also the level or range that the term implies. In the case of the CS field, because the CS2013 is considered to best reflect the present terminology, it is most appropriate to construct the terminology dictionary based on CS2013.

## 4. Terminology Extraction System

This section is composed of the development of a terminology extraction system, evaluation of the system, and presentation of the curriculum developed using the system. To achieve this objective,

first, we built a terminology dictionary using CS2013. Second, we developed a terminology extraction system that can extract complex words using the terminology dictionary. Third, we analyzed the high school CS curricula in India and Israel using the system. Based on the analyzed result, we completed a new CS-based content system.

*4.1. Construction of a Terminology Dictionary*

As the terminology dictionary of various fields is presented, there is an open library or ontology related to the developed terminology. However, there is a limit to expressing the structure related to the establishment of curricula considering the specific academic viewpoint and connection with elementary and secondary school. This is because they should be commonly used in various school levels without needing to account for expressions or terms that reflect the characteristics of the curriculum [28]. In order to construct a specialized terminology dictionary for the CS field, it is necessary to consider the terminology that can be included in each school level. The procedure for constructing a terminology dictionary suitable for the above purpose is shown in Figure 2.

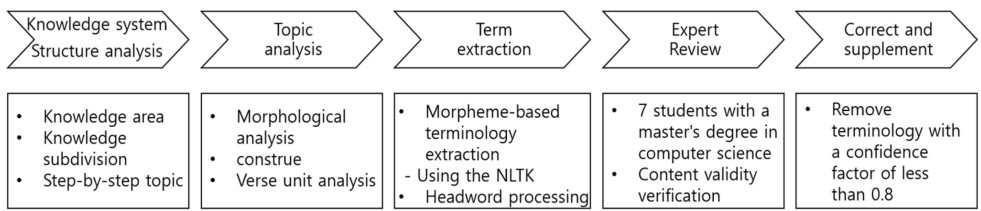

**Figure 2.** Procedure for developing the terminology dictionary.

Step 1) CS2013 is composed of knowledge areas, detailed areas of knowledge, and topics by level. The tiered topics are 'Tier 1', which deals with basic concepts at an introductory level, 'Tier 2', which covers major concepts, and 'Elective', which deals with contents at a deeper level than that of Tier 2. The composition of CS2013 is shown in Figure 3.

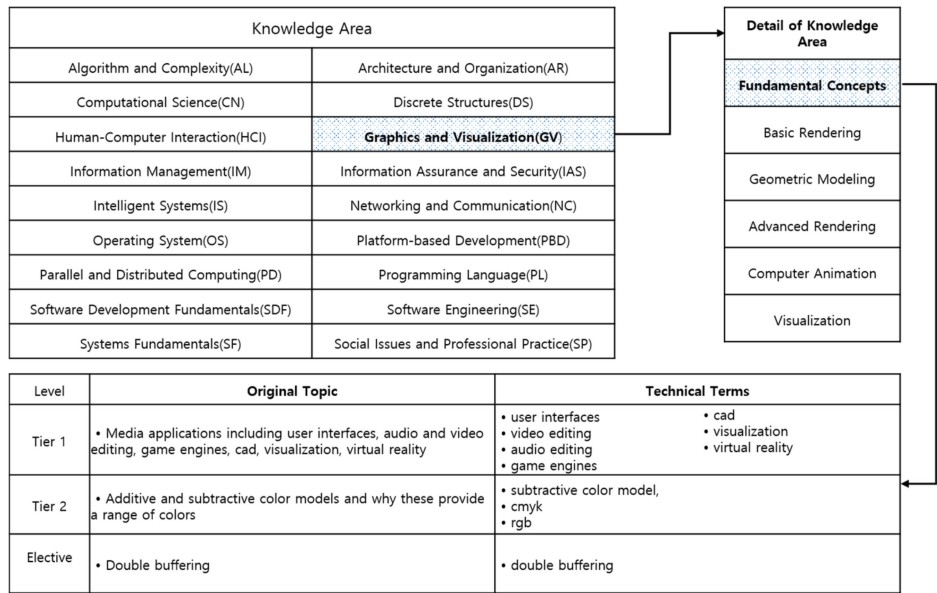

**Figure 3.** Procedure for developing the terminology dictionary.

Step 2) We analyzed the morphological structure, morphemes, and verses (noun phrase, verb phrase, adverb) of the subjects. The results show that the morpheme composition of the jargon in the curriculum has five types.

- Adjectives + singular nouns;
- Singular nouns + singular nouns;
- Adjectives + singular nouns + singular nouns;
- Present participle + singular noun;
- Singular nouns + singular nouns + singular nouns.

That is, 'Tier 1', which deals with basic concepts at an introductory level, as in "Topic (sentence) count of CS2013" in Table 2, consists of 225 sentences. 'Elective', which deals with contents at a deeper level than that of Tier 2, is a total of 623 sentences.

**Table 2.** Number of technical terms in the terminology dictionary extracted from CS2013.

| KA | Topic (Sentence) Count of CS2013 | | | | Technical Term Count of Dictionary | | | |
|---|---|---|---|---|---|---|---|---|
| | Tier1 | Tier2 | Elective | Total | Tier1 | Tier2 | Elective | Total |
| AL | 21 | 14 | 29 | 64 | 94(+73) | 71(+57) | 134(+105) | 299(+235) |
| AR | 0 | 39 | 16 | 55 | 0(+0) | 176(+137) | 59(+43) | 235(+180) |
| CN | 5 | 0 | 27 | 32 | 18(+13) | 0(+0) | 315(+288) | 333(+301) |
| DS | 35 | 5 | 0 | 40 | 149(+114) | 10(+5) | 0(+0) | 159(+119) |
| GV | 4 | 3 | 58 | 65 | 42(+38) | 20(+17) | 250(+192) | 312(+247) |
| HCI | 10 | 8 | 58 | 76 | 69(+59) | 32(+24) | 327(+269) | 428(+352) |
| IAS | 17 | 19 | 65 | 101 | 62(+45) | 110(+91) | 301(+236) | 473(+372) |
| IM | 4 | 16 | 78 | 98 | 17(+13) | 54(+38) | 323(+245) | 394(+296) |
| IS | 0 | 21 | 62 | 83 | 0(+0) | 99(+78) | 312(+250) | 411(+328) |
| NC | 10 | 22 | 4 | 36 | 62(+52) | 55(+33) | 9(+5) | 126(+90) |
| OS | 12 | 19 | 32 | 63 | 50(+38) | 62(+43) | 110(+78) | 222(+159) |
| PBD | 0 | 0 | 21 | 21 | 0(+0) | 0(+0) | 67(+46) | 67(+46) |
| PD | 10 | 12 | 33 | 55 | 47(+37) | 101(+89) | 198(+165) | 346(+291) |
| PL | 11 | 20 | 60 | 91 | 90(+79) | 189(+169) | 245(+185) | 524(+433) |
| SDF | 24 | 0 | 0 | 24 | 144(+120) | 0(+0) | 0(+0) | 144(+120) |
| SE | 9 | 36 | 40 | 85 | 64(+55) | 228(+192) | 198(+158) | 490(+405) |
| SF | 24 | 16 | 6 | 46 | 138(+114) | 72(+56) | 30(+24) | 240(+194) |
| SP | 29 | 14 | 34 | 77 | 140(+111) | 61(+47) | 170(+136) | 371(+294) |
| Total | 225 | 264 | 623 | 1112 | 1186(+961) | 1340(+1076) | 3048(+2425) | 5574(+4462) |

Step 3) In addition to the morpheme type, expressions were extracted that are frequently used in the literature but are not jargon. The extracted terms were normalized by lowercase capitalization and lemmatization.

Step 4) Seven graduate student researchers (five CS majors and two computer education majors) conducted validation of the content of the extracted results.

Step 5) The inter-rater reliability was calculated, and reliability coefficients less than or equal to 0.8 were removed. A more conservative coefficient was used based on the interpretation of high correlation if the correlation coefficient was 0.7 or higher.

Table 2 shows the results of the above procedure.

As shown in Table 2, the total number of topic (sentence) terms of the AL(Algorithm and Complexity) area in Tier 1 of CS2013 was 21. The sentences consisted of 94 technical terms. The size of terminology dictionary developed in this study was 1186, which was about 5.3 times as many. A total of 1340 Tier 2 and 3048 Elective had similar dictionary sizes relative to the total, based on the curriculum theme.

## 4.2. Development of the Terminology Extraction System

This system was developed through Python 3.4 in the Linux 16.04 operating system environment. The terminology extraction system should be able to process the inputted curriculum documents based on the terminology dictionary. In the curriculum, the four stages of terminology extraction are preprocessing, data structuring, analysis, and summary of results. Figure 4 shows the terminology extraction system model implemented in this study.

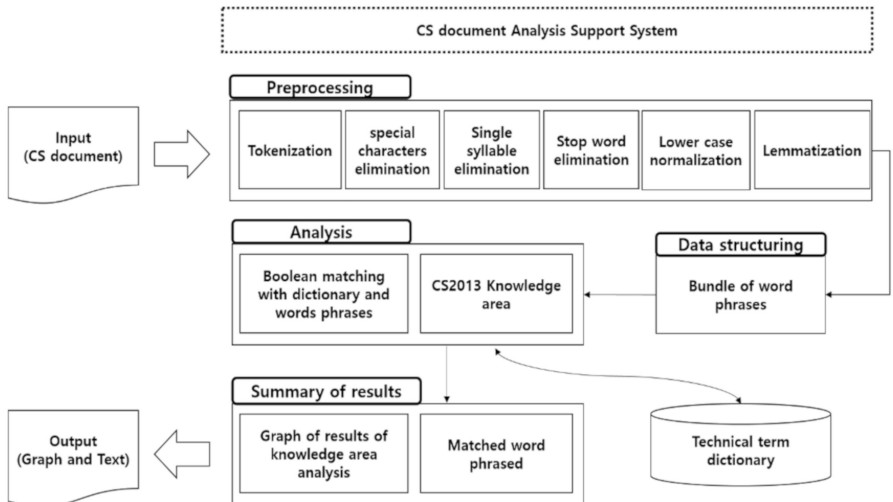

**Figure 4.** Proposed CS document analysis support system.

The preprocessing step translates the inputted curriculum documents into a form suitable for the purpose of analysis [28]. This study used the wordnet nltk module for natural language processing. The preprocessing consists of six steps: tokenization, elimination of special characters, removal of one syllable, elimination of idioms, lowercase normalization, and lemmatization. We split the words into tokens and removed the special characters to improve the matching between terms and the precision of the search. We removed letters and abbreviations denoted by one syllable that did not affect the meaning, and converted all the terms in the document to lowercase. Finally, preprocessing was completed through a lemmatization processing, which normalized various types of terms into a basic dictionary form. Lemmatization can improve the accuracy of a search by preserving part-of-speech information [29].

In the data structuring step, N-grams (i.e., unigrams, bigrams, and trigrams) are applied in units of words as shown in Figure 5. After extracting the first n length words from the topic, the process of extracting n lengths by moving to the next word iterates until the end of the main word.

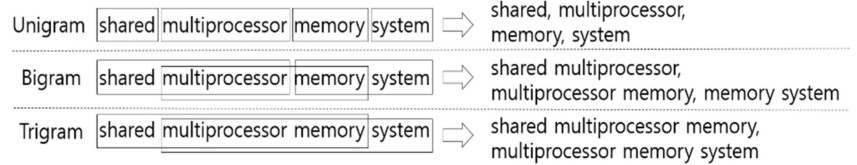

**Figure 5.** N-gram procedure.

As a result of the entire content analysis of CS2013, the maximum value of n was extracted as 3 because the maximum terminology length was shown to be less than 3 words. Despite the complexity of redundancy, all cases were extracted to improve the extraction possibility of compound nouns.

The analysis step is a process of matching the structured data based on the CS2013-based terminology dictionary. In other words, it is a process of analyzing which knowledge domain the

terms of the curriculum to analyze are based on the 18 knowledge domains of CS2013. Figure 6 is an algorithm for a specific procedure for matching term extraction.

```
Algorithm : Dictionary based Matching algorithm
         Input : SD /*the structured data list*/
         Output : ResultMT/*the matched Topic list */,
                TC/*the matched Topic count list in Detail of Knowledge area */
         Definition: CSDictionary /* the dictionary of CS2013 */,
                     KA /* Knowledge area*/,
                     DKA /* Detail of Knowledge area   */,
                     TDKA /* topic in Detail of Knowledge area   */
1        begin
2          Initialize ResultMT = [], TC = []
3          for each i in len(KA)
4            for each j in len(DKA)
5              for each k in 3 // looping for Tier1, Tier2, Elective
6                flag = False
7                for each l in len(TDKA)
8                  for each m in len(SD)
9                    if(BooleanRetrievals(CSDictionary[i][j][k][l], SD[m]) == True)
10                       Save(ResultMT, SD[m]) // save sd[m] in ResultTT list
11                         flag = True
12                   end
13                 end
14               end
15               if flag == True
16                 TC[i][j][k] = TC[i][j][k] + 1
17               end
18             end
19           end
20         end
21         return ResultMT, TC
22       end
```

**Figure 6.** Algorithm for the analysis procedure.

The structured data set is compared with the CS2013 terminology dictionary. The included items are each of the 18 knowledge areas (KA), 136 detailed areas of knowledge areas (KDA), 1113 tiered (Tier 1, Tier 2, and Elective) topics, and restructured detailed topics of each knowledge area

If the data of the structured data list (SD) is retrieved from the terminology dictionary (CSDictionary), the index of the data and the retrieved position are increased. For example, if the data of a search target is found in a terminology dictionary, it is checked to see which of the 18 areas the data corresponds to. When the structured dataset searches all the areas of the terminology dictionary, it returns the retrieved data list (ResultMT) and the location index list (TC).

*4.3. System Application and Evaluation*

While there are no major differences in the goals of the respective education systems of each country, the interdisciplinary system has different forms because it takes into account the national social context, historical context, and educational system. We analyzed two curriculum documents using the terminology extraction system developed in this study. That is, we used the CS curriculum documents of India and Israel are used as input data of the terminology extraction system to verify which knowledge area of CS2013 matches the terms used in the curriculum of the two countries. The reason why we selected the respective curricula of India and Israel as the targets of extraction is because the purpose and learning elements presented in their curricula are concrete, unlike other countries.

Table 3 compares the content extracted from the CS2013 terminology dictionary and the curricula of India and Israel.

**Table 3.** Number of technical terms extracted from the curriculum.

| KA | CS2013 | | | India | | | Israel | | |
|---|---|---|---|---|---|---|---|---|---|
| | Tier1 | Tier2 | Elective | Tier1 | Tier2 | Elective | Tier1 | Tier2 | Elective |
| AL | 21(9.3) | 14(5.3) | 29(4.7) | 4(5.6) | 2(2.9) | 4(3.4) | 6(8.8) | 2(3.2) | 4(4.2) |
| AR | 0(0) | 40(15.1) | 16(2.6) | 0(0) | 11(16.2) | 1(0.9) | 0(0) | 13(21) | 0(0) |
| CN | 5(2.2) | 0(0) | 27(4.3) | 2(2.8) | 0(0) | 16(13.7) | 2(2.9) | 0(0) | 19(19.8) |
| DS | 35(15.6) | 5(1.9) | 0(0) | 5(6.9) | 1(1.5) | 0(0) | 5(7.4) | 0(0) | 0(0) |
| GV | 4(1.8) | 3(1.1) | 58(9.3) | 3(4.2) | 1(1.5) | 4(3.4) | 2(2.9) | 1(1.6) | 10(10.4) |
| HCI | 10(4.4) | 8(3) | 58(9.3) | 6(8.3) | 3(4.4) | 16(13.7) | 5(7.4) | 3(4.8) | 9(9.4) |
| IAS | 17(7.6) | 19(7.2) | 65(10.4) | 2(2.8) | 6(8.8) | 5(4.3) | 4(5.9) | 4(6.5) | 1(1) |
| IM | 4(1.8) | 16(6) | 78(12.5) | 1(1.4) | 3(4.4) | 25(21.4) | 1(1.5) | 3(4.8) | 13(13.5) |
| IS | 0(0) | 21(7.9) | 62(10) | 0(0) | 4(5.9) | 7(6) | 0(0) | 3(4.8) | 6(6.3) |
| NC | 10(4.4) | 22(8.3) | 4(0.6) | 8(11.1) | 4(5.9) | 0(0) | 2(2.9) | 1(1.6) | 0(0) |
| OS | 12(5.3) | 19(7.2) | 32(5.1) | 6(8.3) | 2(2.9) | 7(6) | 2(2.9) | 2(3.2) | 7(7.3) |
| PBD | 0(0) | 0(0) | 21(3.4) | 0(0) | 0(0) | 3(2.6) | 0(0) | 0(0) | 0(0) |
| PD | 10(4.4) | 12(4.5) | 33(5.3) | 0(0) | 0(0) | 0(0) | 2(2.9) | 2(3.2) | 5(5.2) |
| PL | 11(4.9) | 20(7.5) | 60(9.6) | 7(9.7) | 13(19.1) | 13(11.1) | 7(10.3) | 10(16.1) | 9(9.4) |
| SDF | 24(10.7) | 0(0) | 0(0) | 11(15.3) | 0(0) | 0(0) | 13(19.1) | 0(0) | 0(0) |
| SE | 9(4) | 36(13.6) | 40(6.4) | 1(1.4) | 10(14.7) | 4(3.4) | 4(5.9) | 12(19.4) | 5(5.2) |
| SF | 24(10.7) | 16(6) | 6(1) | 6(8.3) | 5(7.4) | 2(1.7) | 5(7.4) | 4(6.5) | 1(1) |
| SP | 29(12.9) | 14(5.3) | 34(5.5) | 10(13.9) | 3(4.4) | 10(8.5) | 8(11.8) | 2(3.2) | 7(7.3) |
| Total | 225(100) | 265(100) | 623(100) | 72(100) | 68(100) | 117(100) | 68(100) | 62(100) | 96(100) |

The results of Tier 1 are as follows. Discrete structures (DSs, 15.6), social issues and professional practice (SP, 12.9), software development fundamentals (SDFs, 10.7), and systems fundamentals (SFs, 10.7) were the top four areas that occupied the highest portion of the knowledge domain in the CS2013-based terminology dictionary. The curriculum in India included a lot of jargon in the order of SDF (15.3), SP (13.9), and networking and communication (NC, 11.1). In Israel, SDF (19.1) and SP (11.8) included the most jargon in the same way as the Indian curriculum, followed by programming languages (PLs, 10.3). The two training courses in India and Israel have been specifically included in two of the top three areas of the Tier 1 of the CS2013 terminology dictionary, with approximately 30% of the terminology associated with SDF and SP.

In Tier 2, India and Israel have the same top three knowledge areas of higher education but in a different order. Of these, architecture and organization (AR) and software engineering (SE) were included in two of the top three areas in Tier 2 of the CS2013 terminology dictionary. In Elective, the three training courses were common, and the information management (IM) terms had a high percentage of terminology. Table 4 shows technical terms extracted from the algorithm and programming domains, with the exception of duplicate technical terms in India and Israel.

**Table 4.** Technical terms extracted from Algorithm and Complexity(AL), programming languages (PLs), and software development fundamentals (SDFs).

| | Extracted technical terms |
|---|---|
| India | variable, assignment, number, character, list, array, average, syntax, min, mode, max, control structure, recursion, pattern matching, argument, function, method, reference, encapsulation, inheritance, constructor, error, sorting, binary search, traversal, run-time |
| Israel | problem-solving, input, processing, output, control structure, data structure, list, graph, square root, variable, assignment, number, character, array, syntax, conditional, concept of recursion, iterative, reference, argument, returning, function, abstract data type, error, halting problem, sorting, merge sort, search, binary tree, traversal, worst case, square root, run-time |

### 4.4. Knowledge Area Composition Using Word2vec

The clustering results of Word2vec are based on the technical terms extracted from the high school CS curricula in India and Israel. This study was limited to knowledge related to programming. The word-embedding results are shown in Figure 7.

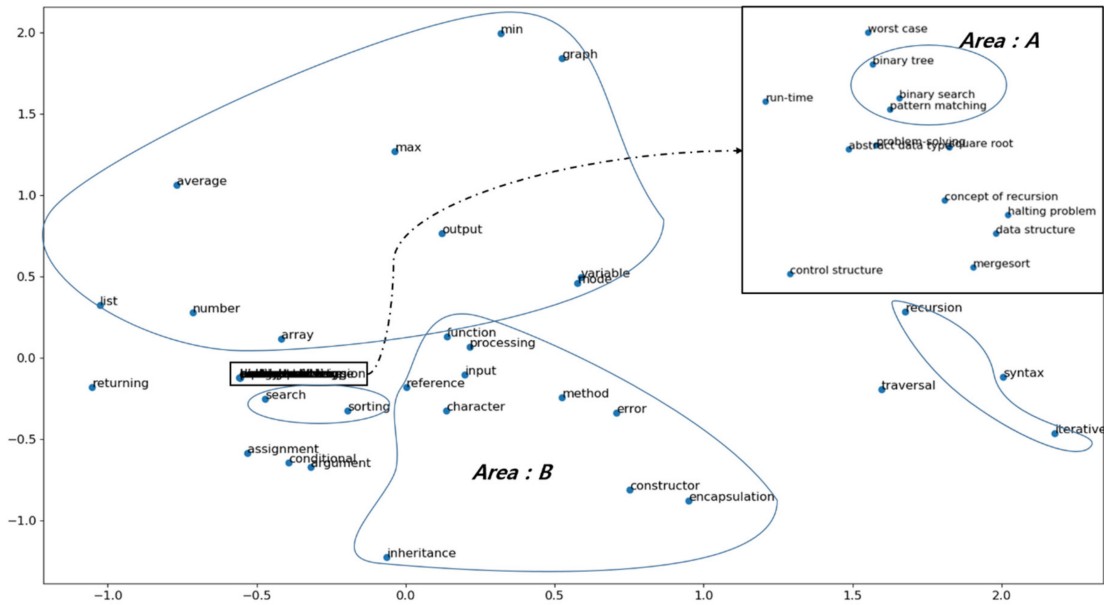

**Figure 7.** Embedding results.

In the case of curriculum, not many terms were utilized, but it can be said that every word was highly significant. The results of Word2Vec were clustered sparsely, but terms could be classified into specific knowledge domains. The clustering results were for PL among the terms of various knowledge areas. For example, in the case of A, binary search, pattern matching, binary tree, control structure, and others constituted a single cluster. In other words, it is interpreted that it could be used as a DKA or element of KA relating to data structures. In the case of B, a cluster is composed of contents closely related to functions such as reference, method, encapsulation, and inheritance, and this fact was referenced in the programming contents structure.

## 5. Curriculum Development

This study develops a system that helps to consider continuity and sequence based on the knowledge system of higher education when developing a CS curriculum. It was applied to the development of high school curricula reflecting not only the CS knowledge system but also the curricula of other countries. Curriculum development consists of three stages.

First, we established a curriculum development strategy based on the clustering results of the word2vec. The result of Word2Vec is because the relationship between the extracted terms is considered very close. Finally, the curriculum is designed for students to complete their own programs.

Second, we organized knowledge areas and topics of the curriculum. A questionnaire was given to determine the validity and suitability of 'knowledge areas: topic' for 12 CS professors. Both the fitness and validity were measured using a 5-point Likert scale, where higher scores indicate appropriateness or relevance and lower scores indicate a lack of suitability and validity. The analysis results of the expert responses are shown in Table 5.

**Table 5.** Response result.

| KA | Topic | Compatibility | Feasibility |
|---|---|---|---|
| Information Society and Information Technology | Information technology development and social change<br>Information technology and problem solving | 4.17 | 4.33 |
| Algorithm Overview | Algorithm Concepts<br>Method of Representation of Algorithms<br>Algorithm design | 4.33 | 4.58 |
| Data structure | Basic data structure<br>- linear data structure<br>- nonlinear data structure | 4.25 | 4.45 |
| Programming Basics | Configuration of Programming<br>I/O function<br>Variable, type, expression, assignment | 4.25 | 4.42 |
| | Selection structure<br>Repeating structure<br>Passing Functions and Parameters | 4.08 | 4.50 |
| | Project | 3.83 | 4.33 |
| Design and Implementation of Algorithms | Sorting algorithm type | 3.83 | 4.33 |
| | Types of search algorithms | 3.67 | 4.25 |
| | Project | 3.83 | 4.25 |
| Algorithm Performance Analysis Basics | Basics of algorithm efficiency analysis<br>Space Complexity and Time Complexity<br>Measure the performance of the algorithm | 3.33 | 3.92 |
| Implementation of algorithm performance analysis | Comparison of efficiency between algorithms | 3.25 | 3.83 |

As a result, the fitness for the domain was more than 3.50, except for the algorithm performance analysis basis and algorithm performance analysis implementation domain. The validity of all areas was over 3.50, and information society, algorithms, and programming areas were higher than 4.00.

Third, we modified the curriculum by referring to the results of the expert feasibility study. Table 6 shows the final curriculum developed through modification according to expert opinions.

'Algorithm Performance Analysis Basics', which showed a low degree of adaptability due to high difficulty of the contents, included only the contents of time complexity. For performance analysis basics and implementation, we reconstructed the approach from the viewpoint of performance in the learning process of algorithm principles rather than constituting it as an independent domain.

**Table 6.** Algorithms and programming curriculum.

| KA | DKA | Topics |
|---|---|---|
| Information Technology Overview | Social influence of information technology | Information technology development and social change<br>Information culture<br>Utilization of information technology |
| Algorithm | Algorithm Overview | Algorithm Concepts<br>Method of Representation of Algorithms<br>Basic data structure |
| | Algorithm design | Sorting algorithm<br>Search algorithm |
| | Analysis of algorithm performance | Algorithm complexity |
| Programming | Programming Basics | Programming environment<br>I / O function<br>Variables, data types, operators |
| | | Selection structure<br>Repeating structure<br>Passing Functions and Parameters |
| | Program implementation | Implementation of sorting algorithm<br>Implementation of search algorithm |

## 6. Conclusions

Society is changing smartly with the rapid development of IT. To cope with these changes, it is necessary to provide students with a curriculum that reflects global trends.

A curriculum is structured according to values that are important to each country in view of the sociocultural context. However, as it is based on the knowledge system, it is necessary to consider the separation of knowledge and functions and the extraction of basic concepts and core ideas. In other words, it is necessary to consider the depth and breadth of education contents based on terminology. The composition of terminology-based curricula can be considered to have a stable frame of content selection. In addition, analyzing the curriculum of each country and referring to the development of the curriculum will help to construct the curriculum to reflect changes over time.

The purpose of this study is to extract technical terms related to CS to help organize a systematic and scientific curriculum and develop a smart system for supporting curriculum development. The framework for extracting terms in the development of the proposed system is CS2013, which is the standard of higher education curriculum for CS. The existing CS terminology dictionary is considered to be limited because it does not include a sufficient variety of expressions or terms to analyze the terminology of the curriculum. We have developed a system that can analyze the terminology of elementary and junior high school CS curriculum and compare and analyze curriculum terms between countries. The developed system has the following characteristics. First, the terminology dictionary is defined to have a hierarchical structure according to the hierarchy of terms, thereby enhancing usability. Second, the objectivity of the system was improved by analyzing higher education curricula to construct the index for elementary and middle schools. Third, the system was designed to be able to select terms by taking into consideration not only the analysis of the curriculum but also aspects for development, thereby improving the efficiency of system utilization. This study contributed to a more professional, objective, and efficient improvement by utilizing computing technology in the field of curriculum analysis and development within the field of education.

**Author Contributions:** Conceptualization, H.W. and J.K.; methodology, H.W.; software, H.W.; validation, H.W. and J.K.; writing—original draft preparation, H.W. and J.K.; writing—review and editing, H.W. and J.K.; visualization, H.W.; project administration, W.L.; funding acquisition, W.L. All authors have read and agreed to the published version of the manuscript.

**Funding:** This work was supported by the National Research Foundation of Korea (NRF) grant funded by the Korea government (MSIP; No. 2016R1A2B4014471).

**Conflicts of Interest:** The authors declare no conflict of interest.

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
