# Peer review of "Development of Curriculum Design Support System Based on Word Embedding and Terminology Extraction"

_electronics, doi:10.3390/electronics9040608_

Round 1
Reviewer 1 Report
the lines of the tables are not clear. I am not sure if it was intended.
According to the ABET accreditation in the USA, topics must include the followings
- Techniques, skills, and tools necessary for computing practice.
- Principles and practices for secure computing.
- Local and global impacts of computing solutions on individuals, organizations, and society.
It would be better if the outcomes of the system are evaluated based on specific criteria like ABET, rather than individual subjective opinion.
The topics in 5. Curriculum Development does not include any tools for KA.
Author Response
1. the lines of the tables are not clear. I am not sure if it was intended.
- The last line in Table 2 is intended to emphasize "Total." However, we revised it according to your suggestion.
2. According to the ABET accreditation in the USA, topics must include the followings
- Techniques, skills, and tools necessary for computing practice.
- Principles and practices for secure computing.
- Local and global impacts of computing solutions on individuals, organizations, and society.
It would be better if the outcomes of the system are evaluated based on specific criteria like ABET, rather than individual subjective opinion.
- Currently, in Korea, ABEEK (ACCREDITED ENGINEERING EDUCATION SYSTEM), which is similar to ABET, is providing certification by presenting educational program standards and guidelines for engineering and related education in universities. ABEEK's certification is based on eight criteria: educational objectives, outcomes, curriculum, students, faculty, educational environment, program improvement, certification criteria by major. The contents related to education are evaluated through the curriculum; however, the specific criteria are about the operation of the overall program, such as the inclusion of lab and research courses and completion of more than 18 credit hours. The specific criteria are the content needed to reach the academic achievement. They do not assess the system of knowledge (contents system), as suggested in this study. Therefore, the contents system is verified by the experts conducting the ABEEK's certification.
3. The topics in section5 Curriculum Development does not include any tools for KA.
- Thank you for your feedback. This study focused on the knowledge aspect of what to teach in a curriculum. In other words, how to teach (in terms of methods or tools) at university could be freely structured in accordance with the instructor’s situation. Since many tools can be presented to teach a single piece of knowledge, we tried to present various tools that can be adjusted for an educational object, subject, and method in consideration of the reviewer's opinion, when this study's program is conducted in the education field later.

Reviewer 2 Report
Abstract..
"We analyzed a higher-education CS standards document and compiled a dictionary of technical terms with a hierarchical clustering structure." Maybe consider "The Authors...
Research Methods - Approach clarity is needed
"In this study, based on 1,112 subjects presented in CS2013, 1,186 Tier 1 terms, 1,340 Tier 2 terms, and 3,048 Elective terms were reorganized. The number of terminology terms in the developed terminology dictionary is as follows. "
Over which year, or a couple years.
This study applied the CBSE Senior School Computer Science curriculum (2015) and Israel’s High School Computer Science Curriculum (1999)
Then suddenly you mentioned - two sets of data from different years? as this skewed your data sets?; This is not clear, where has this first data set come from? was the 1186 came from 1999 or the 1340 second term in 2015. or was there 4 sets of data? one from 1999 and another from 2015? questions to be answered?
Figure 7. Embedding results - is very unclear..
References - please pick a suitable reference approach as it is a mess? you have Songchang Jin et al.,...
then, HoSung. Woo, in another. This is not writing in an academic style at all?
Author Response
1. "We analyzed a higher-education CS standards document and compiled a dictionary of technical terms with a hierarchical clustering structure." Maybe consider "The Authors...
- I revised it into author according to your suggestion.
2. Research Methods - Approach clarity is needed.
"In this study, based on 1,112 subjects presented in CS2013, 1,186 Tier 1 terms, 1,340 Tier 2 terms, and 3,048 Elective terms were reorganized. The number of terminology terms in the developed terminology dictionary is as follows. "
Over which year, or a couple years.
This study applied the CBSE Senior School Computer Science curriculum (2015) and Israel’s High School Computer Science Curriculum (1999)
Then suddenly you mentioned - two sets of data from different years? as this skewed your data sets?; This is not clear, where has this first data set come from? was the 1186 came from 1999 or the 1340 second term in 2015. or was there 4 sets of data? one from 1999 and another from 2015? questions to be answered?
- I apologize for the confusion due to the lack of explanation on procedures. According to your suggestion, the following revision was made.
- This study was conducted to build a terminology dictionary using CS2013 and develop a terminology extraction system based on the terminology dictionary. In other words, this terminology dictionary was built to provide a reference for constructing a curriculum or searching for knowledge in the CS field.
- This study developed a terminology extraction system and analyzed the curriculum in India and Israel using the developed system. In other words, we tried to find which knowledge appears often in the curriculum documents presented by the two countries at the national level.
- To explain the above contents:
- We described the contents on the procedure in the terminology dictionary construction section of 4.1 more clearly. Through the description of TABLE 2, we presented that the reference of this study is the terminology dictionary built based on CS 2013. Also, we minimized confusion in the description by concisely explaining the system application and evaluation in 4.3.
- For your information, Israel is using the curriculum presented in 1999 without any modifications until now. While the changes in the textbooks are partially made, curriculum documents have not been changed. Although they are old, we used the curriculum documents.
3. Figure 7. Embedding results - is very unclear..
- We have made the following revisions according to your suggestions.
- The results of Word2Vec were clustered sparsely, but terms could be classified into specific knowledge domains. For example, in the case of A, binary search, pattern matching, binary tree, control structure, and others constituted a single cluster. In other words, it is interpreted that it could be used as a DKA or element of KA relating to data structures. In the case of B, a cluster is composed of contents closely related to functions such as reference, method, encapsulation, and inheritance, and this fact was referenced in the programming contents structure.
- In the case of curriculum, not many terms are utilized, but it can be said that every word is highly significant. Therefore, although the results of Word2Vec are sparse, we presented the above interpretation because the relationship between the extracted terms are considered to be very close.
4. References - please pick a suitable reference approach as it is a mess? you have Songchang Jin et al.,...
then, HoSung. Woo, in another. This is not writing in an academic style at all?
- We have revised the reference format in accordance with your suggestion.
